# Using Style Ambiguity Loss to Improve Aesthetics of Diffusion Models

## Abstract

Teaching text-to-image models to be creative involves using style ambiguity loss. In this work, we explore using the style ambiguity training objective, used to approximate creativity, on a diffusion model. We then experiment with forms of style ambiguity loss that do not require training a classifier or a labeled dataset, and find that the models trained with style ambiguity loss can generate better images than the baseline diffusion models and GANs.

## 1 Introduction

With every new invention comes a new wave of possibilities. Humans have been making pictures since before recorded history, so its only natural that there would be interest in computational image generation. Artificially generating photographs that are indistinguishable from real ones has become so easy and effective that there is even concern over "deepfakes" being used for propaganda or illicit purposes (Pawelec, 2022). However, there is also demand for machine-generated images that are not just realistic but artistic, as exemplified by the picture that sold for nearly half a million dollars (Christie's, 2018) at auction, or the picture used in the sitcom *Silicon Valley* (AICAN, 2024). While the definition of art is somewhat philosophical and beyond the scope of this paper, it most certainly has to be creative. Creative assets are usually defined as being "novel and useful" (Diedrich et al., 2015). A string of random characters is novel in that we cannot predict the next characters. However, if the purpose of characters is to compose words that compose sentences that communicate a message, then a string of random characters is not useful. Most people would not consider a string of random characters to be creative or art. If the utility of an image is to depict objects and scenes that humans understand and can recognize, then generative models perform well in that regard. Novelty, on the other hand has been underexplored. A breakthrough was the invention of the Creative Adversarial Network (Elgammal et al., 2017), which used a style ambiguity loss to train a generator network to generate images that could not be classified as belonging to a particular style. However, this style ambiguity loss requires a pretrained classifier: Every set of styles or concepts requires training a classifier before even training a model to generate images, which itself takes time and requires a labeled dataset, the collection and curation of which can be expensive. To circumvent these issues, we propose using a classifier for style ambiguity that does not require any additional training and can be easily applied to any dataset, labeled or unlabeled. Additionally, the Creative Adversarial Network was based off of the GAN framework, which have fallen out of favor compared to the more powerful diffusion models (Luo, 2022), so we propose training a diffusion model with style ambiguity loss. Our contributions are as follows:

- We applied style ambiguity loss to diffusion models via reinforcement learning

- We developed versatile CLIP-based and K-Means-based creative style ambiguity losses that do not require training a separate style classifier on a labeled dataset.

- Empirically, we find that training a diffusion model with style ambiguity loss teaches the model to generate very novel outputs that may score higher on empirical metrics than diffusion models trained without style ambiguity loss and GANs trained with style ambiguity loss. We use both automated methods and user studies as metrics.

## 2 Related Work

### 2.1 Creativity

Creative work has been formulated as work having novelty, in that it differs from other similar objects, and also utility, in that it still performs a function (Cropley, 2006). For example, a Corinthian column has elaborate, interesting, unexpected adornments (novelty) but still holds up a building (utility). A distinction can also be made between "P-creativity", where the work is novel to the creator, and "H-creativity" where the work is novel to everyone (Boden, 1990). Computational techniques to be creative include using genetic algorithms (DiPaola & Gabora, 2008), reconstructing artifacts from novel collections of attributes (Iqbal et al., 2016), and most relevantly to this work, using Generative Adversarial Networks (Elgammal et al., 2017) with a style ambiguity loss.

### 2.2 Computational Art

One of the first algorithmic approaches dates back to the 1970s with the now primitive AARON (McCorduck, 1991), which was initially only capable of drawing black and white sketches. Generative Adversarial Networks (Goodfellow et al., 2014), or GANs, were some of the first models to be able to create complex, photorealistic images and seemed to have potential to be able to make art. Despite many problems with GANs, such as mode collapse and unstable training (Saxena & Cao, 2023), GANs and further improvements (Arjovsky et al., 2017; Karras et al., 2019; 2018) were state of the art until the introduction of diffusion Sohl-Dickstein et al. (2015). Diffusion models such as IMAGEN (Saharia et al., 2022) and DALLE-3 (Betker et al.) have attained widespread commercial success (and controversy) due to their widespread adoption.

### 2.3 Reinforcement Learning

Reinforcement learning (RL) is a method of training a model by having it take actions that generate a reward signal and change the environment, thus changing the impact and availability of future actions Qiang & Zhongli (2011). RL has been used for tasks as diverse as playing board games (Silver et al., 2017), protein design (Lutz et al., 2023), self-driving vehicles (Kiran et al., 2021) and quantitative finance (Sahu et al., 2023). Policy-gradient RL (Sutton et al., 1999) optimizes a policy $\pi$ that chooses which action to take at any given timestep, as opposed to value-based methods that may use a heuristic to determine the optimal choice. Examples of policy gradient methods include Soft Actor Critic (Haarnoja et al., 2018), Deep Deterministic Policy Gradient (Lillicrap et al., 2019) and Trust Region Policy Optimization (Schulman et al., 2017a).

## 3 Background

### 3.1 Creative Adversarial Network

A Generative Adversarial Network, or GAN (Goodfellow et al., 2014), consists of two models, a generator and a discriminator. The generator generates samples from noise, and the discriminator detects if the samples are drawn from the real data or generated. During training, the generator is trained to trick the discriminator into classifying generated images as real, and the discriminator is trained to classify images correctly. Given a generator $G : \mathbb{R}^{noise} \to \mathbb{R}^{h \times w \times 3}$, a discriminator $D : \mathbb{R}^{h \times w \times 3} \to [0, 1]$ real images $x \in \mathbb{R}^{h \times w \times 3}$, and noise $\mathcal{Z} \in \mathbb{R}^{noise}$, the objective is:

$$\min_G \max_D \mathbb{E}_x[log(D(x)] + \mathbb{E}_{\mathcal{Z}}[log(1 - D(G(\mathcal{Z})))]$$

Elgammal et al. (2017) introduced the Creative Adversarial Network, or CAN, which was a DCGAN (Radford et al., 2016) where the discriminator was also trained to classify real samples, minimizing the style classification loss. Given $N$ classes of image (such as ukiyo-e, baroque, impressionism, etc.), the classification modules of the Discriminator $D_C : \mathbb{R}^{h \times w \times 3} \to \mathbb{R}^N$ that returns a probability distribution over the $N_s$ style classes for an image and the real labels $\ell \in \mathbb{R}^N$, the style classification loss was:

$$L_{SL} = \mathbb{E}_{x,\ell}[\mathbf{CE}(D_C(x), \ell)]$$

Where **CE** is the cross entropy function.

The generator was also trained to generate samples that could not be easily classified as belonging to one class. This stylistic ambiguity is a proxy for creativity or novelty. Given a vector $U \in \mathbb{R}^N$, where each entry $u_1, u_2, , , u_N = \frac{1}{N}$, and the classification modules of the discriminator $D_C$ the style ambiguity loss is:

$$L_{SA} = \mathbb{E}_{\mathcal{Z}}[\mathbf{CE}(C(G(\mathcal{Z})), U)]$$

The discriminator was additionally trained to minimize $L_{SL}$ and the generator was additionally trained to minimize $L_{SA}$.

## 3.2 Diffusion

A diffusion model aims to learn to iteratively remove the noise from a corrupted sample to restore the original. Starting with $x_0$, the forward process $q$ iteratively adds Gaussian noise to produce the noised version $x_T$, using a noise schedule $\beta_1 ... \beta_T$, which can be learned or manually set as a hyperparameter:

$$q(x_t|x_{t-1}) = \mathcal{N}(x_t; \sqrt{1-\beta_t}x_{t-1}, \beta_t\mathbf{I})$$

More importantly, we also want to model the reverse process $p$, that turns a noisy sample $x_T$ back into $x_0$, conditioned on some context $c$. As $x_T$ is the fully noised version, $p(x_T|c) = \mathcal{N}(x_T; \mathbf{0}, \mathbf{I})$

$$p_\theta(x_{t-1}|x_t, c) = \mathcal{N}(x_{t-1}; \mu_\theta(x_t, t, c), \Sigma_\theta(x_t, t, c))$$

Once the model has been trained, the reverse process, aka inference, generates a sample from noise $x_T \sim \mathcal{N}(0, 1)$. We use the DDIM technique (Song et al., 2022) for sampling. In this work, we use a variant of diffusion known as Stable Diffusion (Rombach et al., 2022), where $x$ is replaced with a latent embedding $\mathcal{E}(x)$, where $\mathcal{E} : \mathbb{R}^{h \times w \times 3} \to \mathbb{R}^{h_z \times w_z \times c_z}$ is a frozen autoencoder (Kingma & Welling, 2022), and $h_z < h, w_z < w, c_z > 3$.

# 4 Methods

## 4.1 Denoising Diffusion Proximal Optimisation

Introduced by Black et al. (2023), Denoising Diffusion Proximal Optimisation, or DDPO, represents the reverse Diffusion Process as a Markov Decision Process (Bellman, 1957). A similar method was also pursued by Fan et al. (2023). Reinforcement learning training was then applied to a pretrained diffusion model. Following Schulman et al. (2017b), Black et al. (2023) also implemented clipping to protect the policy gradient $\nabla_\theta \mathcal{J}_{DDRL}$ from excessively large updates, and per prompt stat tracking to normalize rewards. We largely follow their method but use a different reward function. We fine-tune off of the pre-existing **stabilityai/stable-diffusion-2-base** checkpoint (Rombach et al., 2022) downloaded from `https://huggingface.co/stabilityai/stable-diffusion-2-base`.

## 4.2 Text Prompts

DDPO does not require any new data, given that we are fine-tuning off of a pretrained checkpoint. However, each time we train the model, we must decide which text prompts to use to condition the generation of images. We used the set of (painting, drawing, art) as our text prompts

## 4.3 Datasets and Labels

Training the CAN, of course, requires a labeled dataset. We used two different real datasets based off of WikiArt (Saleh & Elgammal, 2015):

1. **Full:** the WikiArt dataset as is. Consists of roughly 80k images.

2. **Mediums:** given the the text prompt set (painting, drawing, art), we used BLIP (Li et al., 2022) to generate captions, and then selected the 10 classes that had the highest fraction of their descriptions containing any of the words in **Mediums**. We then used the images in WikiArt that belonged to those classes. Consists of roughly 20k images.

This also meant we had two different sets of style class labels: $L_{full}$, all 27 class labels, used with **Full** and $L_{med}$, the 10 labels used in **Mediums**. For implementation details regarding the captions, and a list of the class labels used for **Mediums** refer to appendix D

## 4.4 DDPO Reward Function

In the original DDPO paper, the authors used four different reward functions for four different tasks. For example, they used a scorer trained on the LAION dataset (Schuhmann & Beaumont, 2022) as the reward function to improve the aesthetic quality of generated outputs. In this paper, we use the reward model based on Elgammal et al. (2017), where the model is rewarded for stylistic ambiguity, combined with a reward for utility. Given a pretrained CLIP (Radford et al., 2021) model, that can return a similarity score for each image-text pair: $CLIP : \mathbb{R}^{text} \times \mathbb{R}^{h \times w \times 3} \to \mathbb{R}$, image $x_0 \in \mathbb{R}^{h \times w \times 3}$ generated with text prompt $s$, cross entropy **CE**, uniform distribution $U \in \mathbb{R}^N$ and a classifier $C : \mathbb{R}^{h \times w \times 3} \to \mathbb{R}^N$ we want to maximize:

$$R(x_0) = -\lambda_{novelty}\mathbf{CE}(C(x_0), U) + \lambda_{utility}CLIP(s, x_0)$$

The first term on the left side of the equation represents style ambiguity loss, and the second term maintains alignment between text and image, essentially keeping the model from straying "too far" from the text prompt; these terms approximate novelty and utility, respectively. We actually have *multiple* choices of classifier, which we dicuss.

### 4.4.1 Discriminator Classifier

We can use the classification module of the CAN discriminator as the classifier in the reward function, setting $C = D_C$. This is the traditional method of style ambiguity loss, and the baseline against which we are comparing the other two types of classifier with. This discriminator was trained at image resolution 512, with batch size 128 without using gradient penalty (Gulrajani et al., 2017). Given that we had two datasets (**Full** and **Mediums**), we actually had to train two of these discriminators, one trained for each.

### 4.4.2 CLIP-Based Style Classifier

For each generated image $x_0$, for each class name $s_i, 1 \leq i \leq N_s$ in the style class label set we want to use, we find $CLIP(s_i, x_0)$. We can then create a vector $(CLIP(s_1, x_0), CLIP(s_2, x_0), , , CLIP(s_{N_s}, x_0))$ and then use softmax to normalize the vector and define the result as $CC(x_0)$. Formally:

$$CC(x_0) = \mathbf{softmax}((CLIP(s_1, x_0), CLIP(s_2, x_0), , , CLIP(s_{N_s}, x_0))$$

Then we set $C = CC$. Given the two sets of labels, $L_{full}$ and $L_{med}$, we had two different types of CLIP-Based classifier.

### 4.4.3 K-Means Image Based Classifiers

Alternatively, $N_I$ source images in a dataset, we can embed the labels or images into the CLIP embedding space $\in \mathbb{R}^{768}$ and perform k-means clustering to generate k centers. Given a CLIP Embedder $E : \mathbb{R}^{h \times w \times 3} \to \mathbb{R}^{768}$ mapping images to embeddings, and the k centers $c_1, c_2, , , c_k$ we can create a vector $(\frac{1}{||E(x_0)-c_1||}, \frac{1}{||E(x_0)-c_2||}, , , , \frac{1}{||E(x_0)-c_k||}$ and then use softmax to normalize the vector and define the result as $KM(x_0)$). Formally:

$$KMEANS(x_0) = \mathbf{softmax}(\frac{1}{||E(x_0) - c_1||}, \frac{1}{||E(x_0) - c_2||}, , , , \frac{1}{||E(x_0) - c_k||})$$

Then we set $C = KM$. Given two datasets (**Full** and **Mediums**), we had two sets of centers.

| Name | $\lambda_{novelty}$ | $\lambda_{utility}$ | Classifier? | Dataset | Inference Steps |
|---|---|---|---|---|---|
| *disc-full* | 1 | 0.25 | Discriminator | Full | 30 |
| *clip-full* | 1 | 0.25 | CLIP-Based | Full | 30 |
| *kmeans-full* | 1 | 0.25 | K-Means Based | Full | 30 |
| *disc-med* | 1 | 0.25 | Discriminator | Mediums | 30 |
| *clip-med* | 1 | 0.25 | CLIP-Based | Mediums | 30 |
| *kmeans-med* | 1 | 0.25 | K-Means Based | Mediums | 30 |
| *utility-30* | 0 | 1 | None | N/A | 30 |
| *utility-10* | 0 | 1 | None | N/A | 10 |
| *basic-30* | 0 | 0 | None | N/A | 30 |
| *basic-10* | 0 | 0 | None | N/A | 10 |

Table 1: Diffusion Methods

## 5 Results

Given three types of classifier and two datasets, we had six models. We refer to those models as the "creative" models. We also wanted to compare these models to a model trained with *just* utility loss and a model that was not trained at all off of the benchmark. For the non-creative models, we also generate samples with a smaller amount of inference steps, to dispel the notion that the "creative" models are just learning blurrier, noisier versions of normal models. A breakdown on all the models is in table 1. Note that for methods *clip-full* and *clip-med*, we don't use the data in the data for the classifier per se; however, we use the style class labels, which are unique to the datasets in question. *clip-full* uses $L_{full}$ and *clip-med* uses $L_{mediums}$.

We generated all images with width and height = 512. The authors used width and height = 256 in the original CAN paper. However, given that larger, more detailed images are preferred by most people, we thought it more relevant to focus on larger images.

### 5.1 Quantitative Evaluation

For each dataset, for each choice of classifier, we an evaluation dataset of 100 images. We used the exact same prompts and random seeds for each. I.e. if the nth image generated by *disc-full* used prompt "painting" and random seed = $z \in \mathbb{Z}$, so would the nth image generated by $M_1, M_2$, etc. We used two scoring metrics to score the models:

- **AVA Score: (AVA)** Consisting of CLIP+Multi-Layer Perceptron (Haykin, 2000), the AVA model was trained on the AVA dataset (Murray et al., 2016) of images and average rankings by human subjects, in order to learn to approximate human preferences given an image. We used the CLIP model weights from the **clip-vit-large-patch14** checkpoint and the Multi-Layer Perceptron weights downloaded from `https://huggingface.co/trl-lib/ddpo-aesthetic-predictor`.

- **Image Reward: (IR)** The image reward model (Xu et al., 2023) was trained to score images given their text description based on a dataset of images and human rankings. We used the **image-reward** python library found at `https://github.com/THUDM/ImageReward/tree/main`.

Results of our experiments are shown in tables 2. Each cell contains the mean (and standard deviation). Evidently, some creative models perform better than the uncreative models; the highest IR score was attained by *disc-full*, and the highest AVA score was attained by *kmeans-med*. However, we note that using fewer labels dramatically improves the IR of the CLIP classifier (comparing *clip-full* to $M4$, and using a smaller dataset improves the IR *and* AVA of the K Means classifier (comparing *kmeans-full* to *kmeans-med*). We see the opposite trend with using a discriminator; *disc-full* scores higher than *disc-med* for both. We further visualize these results in table 3. We see that IR tends to be rather skewed.

| Model | AVA | IR |
|---|---|---|
| *disc-full* | 5.49 ( 0.47 ) | 1.33 ( 0.34 ) |
| *clip-full* | 5.49 ( 0.38 ) | 0.84 ( 1.06 ) |
| *kmeans-full* | 4.98 ( 0.4 ) | -0.88 ( 0.31 ) |
| *disc-med* | 5.12 ( 0.31 ) | -0.84 ( 0.31 ) |
| *clip-med* | 5.39 ( 0.38 ) | 1.08 ( 0.87 ) |
| *kmeans-med* | 5.89 ( 0.39 ) | 1.27 ( 0.37 ) |
| *utility-30* | 5.12 ( 0.29 ) | 0.85 ( 0.87 ) |
| *utility-10* | 5.18 ( 0.33 ) | 0.73 ( 0.78 ) |
| *basic-30* | 4.12 ( 0.88 ) | -1.87 ( 0.58 ) |
| *basic-10* | 4.22 ( 0.84 ) | -2.07 ( 0.43 ) |

Table 2: Aesthetic Scores, Diffusion Models

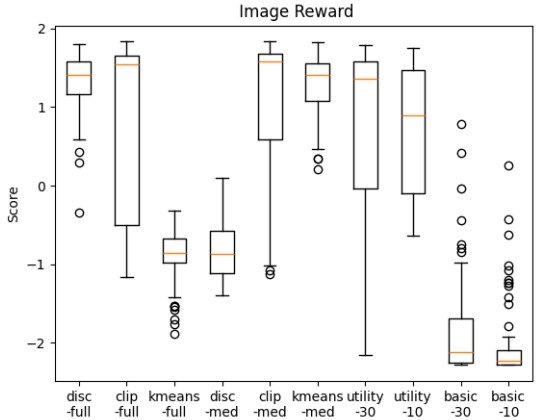 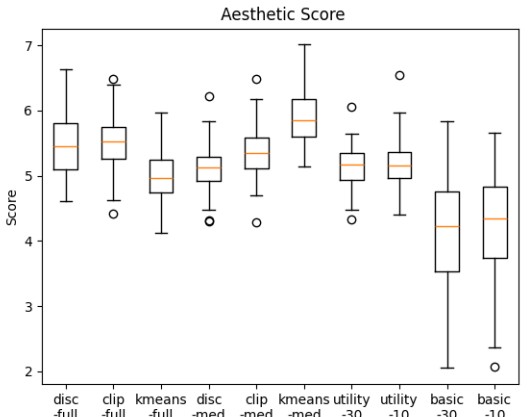

Table 3: Score Box Plots

### 5.1.1 Comparison with GAN Baseline

GANs are largely out of fashion, but we nonetheless thought it worthwhile to test a few variations of the Creative Adversarial Network, as described in Elgammal et al. (2017). Lacking an official repo or weights, we had to base our implementation off of what was described in the paper. The diffusion models *disc-full* and *disc-med* used the classification heads of discriminators trained in this fashion for their reward model, but the images being evaluated in this section were created by the generators used in the Creative Adversarial Network. In the original CAN paper, all images were generated at resolution 256, trained on the **Full** wikiart dataset, with batch size 128 without gradient penalty. However, we wanted to slightly expand our search space, so we tried every possible combination of four choices

1. Image Dimensionality: 256 or 512

2. Dataset: **Full** or **Mediums**

3. Batch Size: 256 or 128

4. Gradient Penalty: GP (We do use GP) or No GP (We do not use GP)

This gave us sixteen combinations, and so we had 16 generators. Scores are shown in table 4. Refer to appendix A for box plots visualizing these results.

The GANs did not perform well. The highest AVA score attained by a GAN was 4.34, which was lower than the AVA score for all of the diffusion models except for *basic-30* and *basic-10*. The highest IR score was -0.89,

| Model | AVA | IR |
|---|---|---|
| Image Dim 256; Full Dataset | | |
| Batch 256 No GP | 3.53 ( 0.13 ) | -2.23 ( 0.02 ) |
| Batch 256 GP | 3.73 ( 0.17 ) | -2.14 ( 0.03 ) |
| Batch 128 No GP | 3.9 ( 0.16 ) | -1.39 ( 0.23 ) |
| Batch 128 GP | 3.79 ( 0.16 ) | -1.29 ( 0.23 ) |
| Image Dim 512; Full Dataset | | |
| Batch 256 No GP | 4.1 ( 0.16 ) | -2.13 ( 0.05 ) |
| Batch 256 GP | 3.64 ( 0.08 ) | -2.2 ( 0.02 ) |
| Batch 128 No GP | 4.25 ( 0.08 ) | -1.78 ( 0.1 ) |
| Batch 128 GP | 4.21 ( 0.14 ) | -0.89 ( 0.1 ) |
| Image Dim 256; Mediums Dataset | | |
| Batch 256 No GP | 3.88 ( 0.19 ) | -1.46 ( 0.18 ) |
| Batch 256 GP | 3.85 ( 0.13 ) | -1.6 ( 0.18 ) |
| Batch 128 No GP | 4.38 ( 0.23 ) | -1.64 ( 0.18 ) |
| Batch 128 GP | 4.04 ( 0.14 ) | -1.39 ( 0.09 ) |
| Image Dim 512; Mediums Dataset | | |
| Batch 256 No GP | 4.31 ( 0.24 ) | -1.53 ( 0.12 ) |
| Batch 256 GP | 3.97 ( 0.14 ) | -1.31 ( 0.12 ) |
| Batch 128 No GP | 4.34 ( 0.2 ) | -1.22 ( 0.13 ) |
| Batch 128 GP | 4.23 ( 0.12 ) | -1.2 ( 0.14 ) |

Table 4: Caption

which was higher than the IR score for both *basic-30* and *basic-10*, and close to (but still worse than) the IR score for *kmeans-full* and *disc-med*. Evidently, GANs do not seem to perform as well as diffusion models for this task. Their inferiority is compounded by the problems of training GANs, such as mode collapse and instability (Arjovsky & Bottou, 2017). The much lower standard deviation of aesthetic scores for the GANs strongly indicates mode collapse.

## 5.2 User Study

We also conducted a user study to evaluate a subset of the models. We chose the 6 creative diffusion models (*disc-full, clip-full, kmeans-full, disc-med* and *kmeans-med*), as well as *utility-30*. Given the high cost of user studies, we had to be judicious in which models we chose to evaluate this way. We omitted *utility-10* given that it's outputs looked a lot like those of *utility-30*. Due to their low aesthetic scores, we also omitted the diffusion models *basic-10* and *basic-30*, and omitted all of the GANs. For each model we did evaluate in this user study, we generated five images with the same seeds. Then, for each image, we asked each user to respond to two questions, based off of the questions used in Elgammal et al. (2017)

1. How much do you like this image: 1-extremely dislike, 2-dislike, 3-Neutral, 4-like, 5-extremely like.

2. Rate the novelty of the image: 1-extremely not novel, 2-some how not novel, 3-neutral, 4 somehow novel, 5-extremely novel.

We used `prolific.com` to find users, who were then redirected to a google forms survey. Each user was paid 12.00 dollars. Median time to complete the survey was 11 minutes and 40 seconds. Results are shown in Table 5. The Subjectivity Mean for each model refers to the average response to question 1 by all users across all images created by that particular model. The Subjectivity Std Dev for each model refers to the average of the standard deviations of the responses to question 1 by all users, across all images created by that particular model. The Novelty Mean for each model refers to the average response to question 2 by all users across all images created by that particular model. The Novelty Std Dev for each model refers to the average of the standard deviations of the responses to question 2 by all users, across all images created by that particular model.

| Model | Subjectivity Mean | Subjectivity Std Dev | Novelty Mean | Novelty Std Dev |
|---|---|---|---|---|
| *disc-full* | 2.94 | 1.151 | 3.06 | 1.079 |
| *clip-full* | 2.28 | 1.227 | 3.12 | 1.255 |
| *kmeans-full* | 2.74 | 0.999 | 2.87 | 1.036 |
| *disc-med* | 3.01 | 1.095 | 2.92 | 1.117 |
| *clip-med* | 2.98 | 1.224 | 2.68 | 1.318 |
| *kmeans-med* | 3.39 | 1.216 | 2.93 | 1.408 |
| *utility-30* | 2.98 | 1.178 | 2.76 | 1.203 |

Table 5: User Study Scores

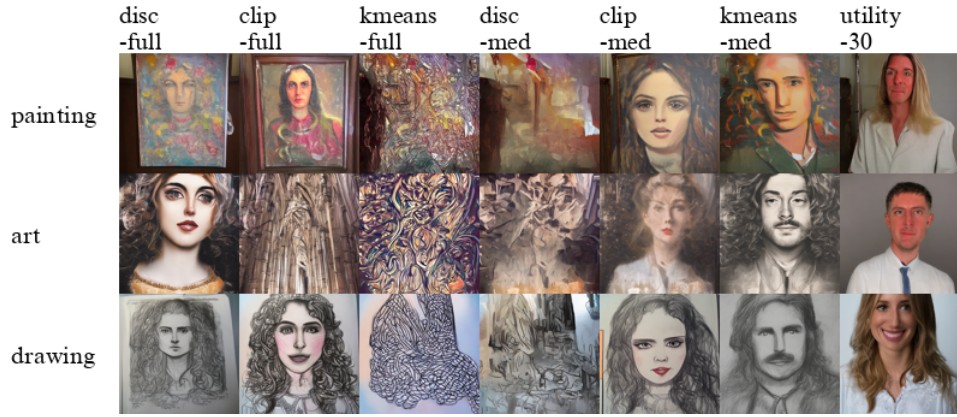

Figure 1: Image Comparisons (Only Creative Models and *utility-30*)

Unsurprisingly, *kmeans-med* has the Subjectivity Mean, given it also had the highest AVA score and the second highest Image Reward score. While *disc-full* had a low subjectivity score, which seemingly contradicts its high score on AVA and Image Reward, it still had the highest Subjectivity Mean of the models trained with the full dataset. The *utility-30* had exactly the median score of the Subjectivity Means. While *clip-full* had the best Novelty Mean, we should note that all models but *clip-med* were more novel than *utility-30*. Thus, this user study demonstrates that our creative loss objectives can improve novelty. While some models such as *disc-full* and *clip-full* sacrifice higher novelty for lower subjective scores, some models like *kmeans-med* and *disc-med* achieve higher novelty *and* higher subjectivity scores than *utility-30*.

### 5.3 Visual Results

We also provide a few visual results in figure 2. Each image in each row was generated with the same prompt and random seed. For more pictures, consult appendix B.

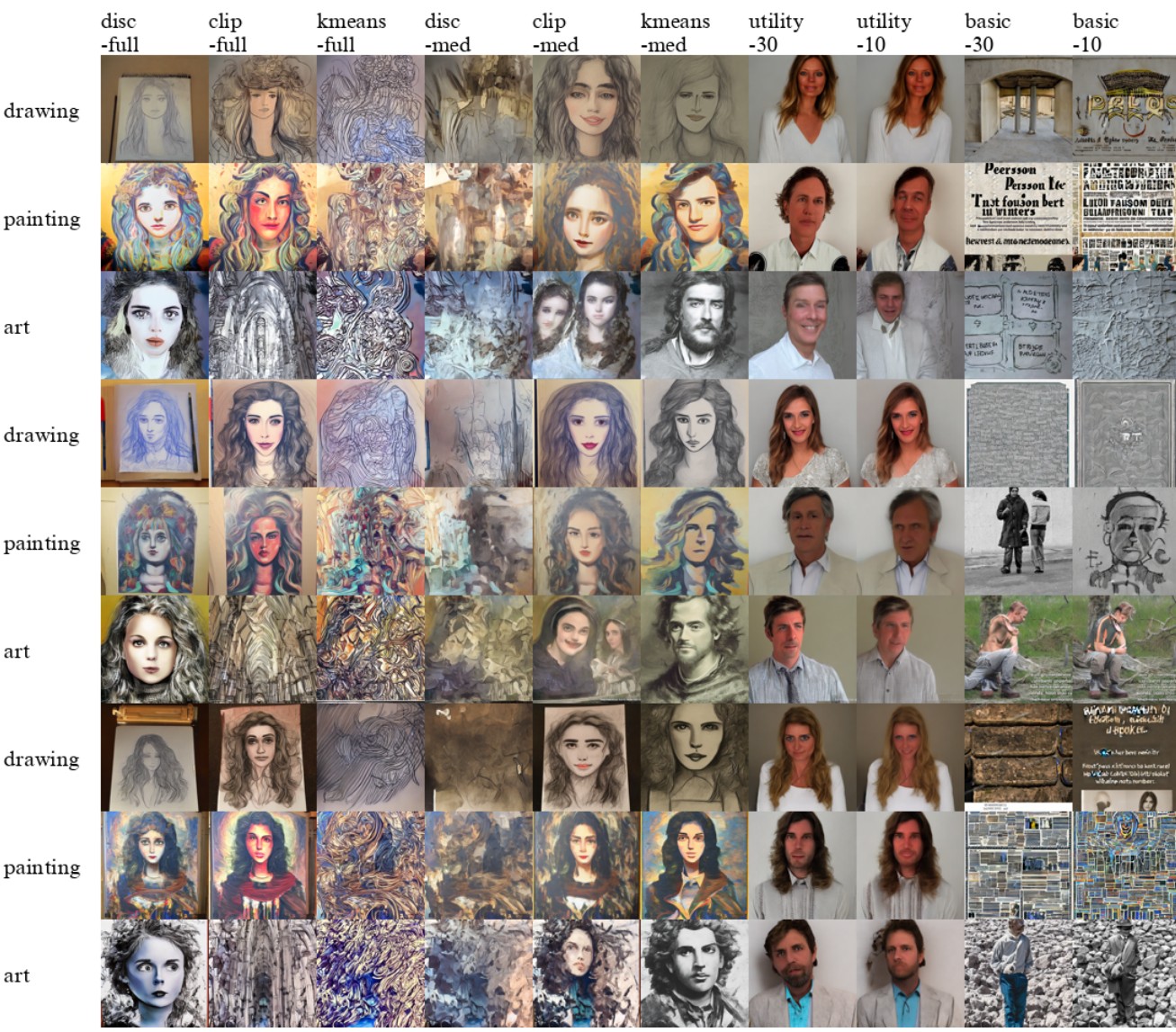

Figure 2: Image Comparisons

Curiously, many of the creative models generate faces. However, given that *utility-30* and *utility-10* do this as well, we suspect that may be a product of the utility function. On the other hand, the *kmeans-full* and *disc-med* models tend to mostly generate abstract noise. This is somewhat similar to the outputs generated in the original CAN paper.

## 5.4 Content and Style Similarities

We also compared the similarities between each model. Given that the nth image generated by each model used the same prompt and seed across all models, we could compare the cosine distance of the embeddings of the nth image for each model with the embeddings of the nth image of every other model, and average them. We used style and content embeddings from the **dino-vits16** checkpoint (Caron et al., 2021) from `https://huggingface.co/facebook/dino-vits16`. Many other works have commented on the ability to extract separable style and content from vision transformers (Tumanyan et al., 2022; Kwon & Ye, 2023).

Unsurprisingly, *utility-30* and *utility-10*, as well as *basic-30* and *basic-10*, have very high similarities, given that they are the same models. The creative models are generally more similar to each other, and more distinct from the uncreative models.

| | disc-full | clip-full | kmeans-full | disc-med | clip-med | kmeans-med | utility-30 | utility-10 | basic-30 | basic-10 |
|---|---|---|---|---|---|---|---|---|---|---|
| disc-full | 1.0 | 0.4 | 0.27 | 0.26 | 0.43 | 0.41 | 0.35 | 0.35 | 0.23 | 0.23 |
| clip-full | 0.4 | 1.0 | 0.31 | 0.29 | 0.43 | 0.39 | 0.34 | 0.34 | 0.22 | 0.23 |
| kmeans-full | 0.27 | 0.31 | 1.0 | 0.4 | 0.26 | 0.24 | 0.18 | 0.19 | 0.22 | 0.24 |
| disc-med | 0.26 | 0.29 | 0.4 | 1.0 | 0.26 | 0.23 | 0.19 | 0.19 | 0.24 | 0.26 |
| clip-med | 0.43 | 0.43 | 0.26 | 0.26 | 1.0 | 0.45 | 0.35 | 0.35 | 0.23 | 0.22 |
| kmeans-med | 0.41 | 0.39 | 0.24 | 0.23 | 0.45 | 1.0 | 0.42 | 0.41 | 0.22 | 0.21 |
| utility-30 | 0.35 | 0.34 | 0.18 | 0.19 | 0.35 | 0.42 | 1.0 | 0.85 | 0.24 | 0.22 |
| utility-10 | 0.35 | 0.34 | 0.19 | 0.19 | 0.35 | 0.41 | 0.85 | 1.0 | 0.24 | 0.22 |
| basic-30 | 0.23 | 0.22 | 0.22 | 0.24 | 0.23 | 0.22 | 0.24 | 0.24 | 1.0 | 0.53 |
| basic-10 | 0.23 | 0.23 | 0.24 | 0.26 | 0.22 | 0.21 | 0.22 | 0.22 | 0.53 | 1.0 |

Table 6: Average Content Similarities

| | disc-full | clip-full | kmeans-full | disc-med | clip-med | kmeans-med | utility-30 | utility-10 | basic-30 | basic-10 |
|---|---|---|---|---|---|---|---|---|---|---|
| disc-full | 1.0 | 0.54 | 0.35 | 0.3 | 0.65 | 0.64 | 0.34 | 0.37 | 0.18 | 0.19 |
| clip-full | 0.54 | 1.0 | 0.41 | 0.35 | 0.58 | 0.54 | 0.33 | 0.35 | 0.18 | 0.19 |
| kmeans-full | 0.35 | 0.41 | 1.0 | 0.41 | 0.29 | 0.3 | 0.12 | 0.13 | 0.14 | 0.18 |
| disc-med | 0.3 | 0.35 | 0.41 | 1.0 | 0.31 | 0.29 | 0.15 | 0.18 | 0.19 | 0.22 |
| clip-med | 0.65 | 0.58 | 0.29 | 0.31 | 1.0 | 0.65 | 0.36 | 0.39 | 0.2 | 0.19 |
| kmeans-med | 0.64 | 0.54 | 0.3 | 0.29 | 0.65 | 1.0 | 0.44 | 0.45 | 0.2 | 0.19 |
| utility-30 | 0.34 | 0.33 | 0.12 | 0.15 | 0.36 | 0.44 | 1.0 | 0.91 | 0.14 | 0.11 |
| utility-10 | 0.37 | 0.35 | 0.13 | 0.18 | 0.39 | 0.45 | 0.91 | 1.0 | 0.15 | 0.12 |
| basic-30 | 0.18 | 0.18 | 0.14 | 0.19 | 0.2 | 0.2 | 0.14 | 0.15 | 1.0 | 0.63 |
| basic-10 | 0.19 | 0.19 | 0.18 | 0.22 | 0.19 | 0.19 | 0.11 | 0.12 | 0.63 | 1.0 |

Table 7: Average Style Similarities

## 5.5 Visualizing The Possibility Space

For each model, we combined each pair of evaluation datasets and then then performed dimensionality reduction using PCA and TSNE to embed images into 2 dimensions. Thus, we could visualize the overlap, or lack thereof, between the possibility space of each model. We see that there are some stark contrasts between the uncreative and creative models, often breaking into very distant clusters.

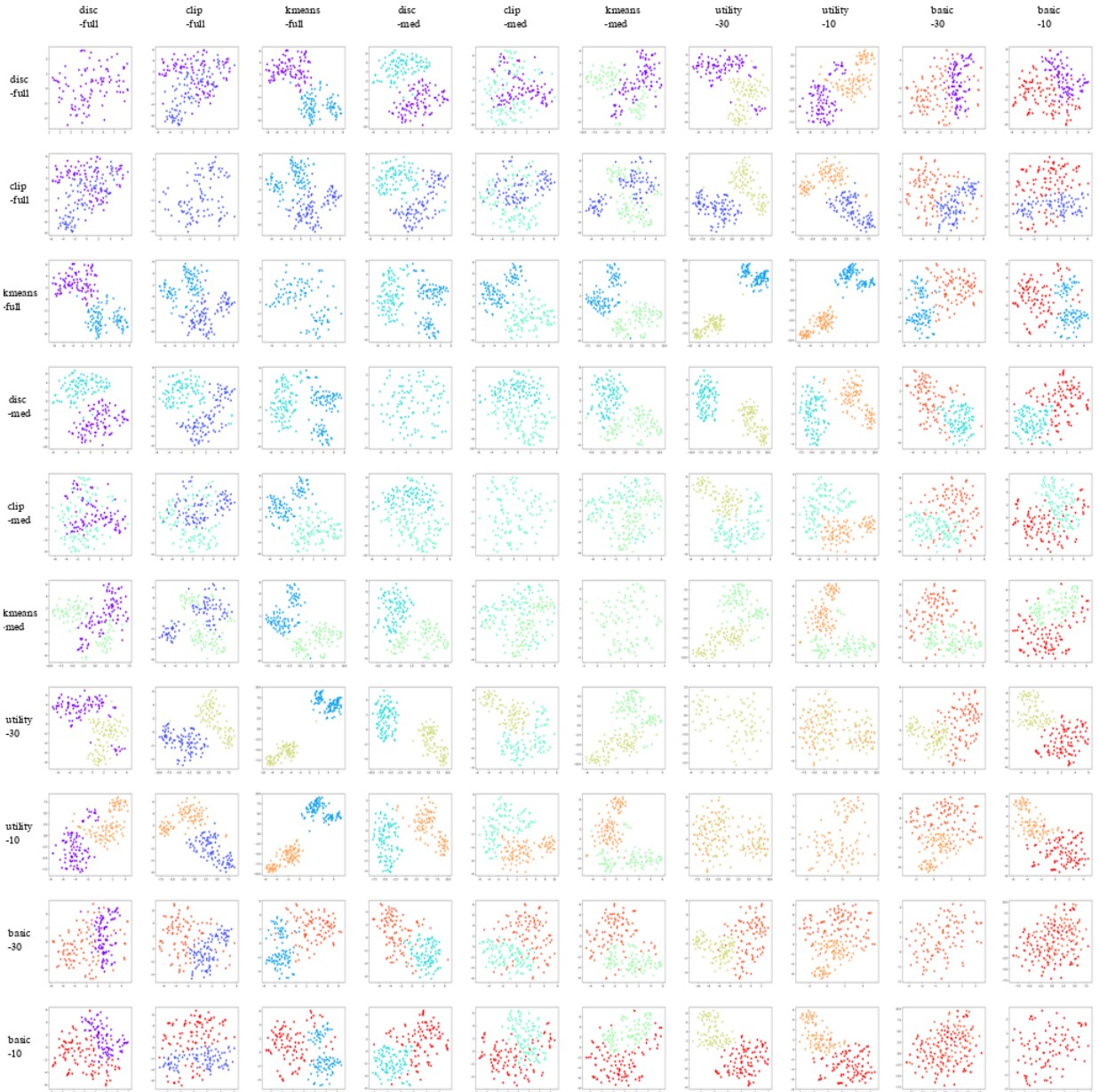

Figure 3: Possibility Space

# 6   Conclusion

In this paper, we introduced innovative techniques to incorporate style ambiguity loss into diffusion models through reinforcement learning. Our primary contributions include the development of versatile CLIP-based and K-Means-based creative style ambiguity losses, which eliminate the need for training a separate style classifier on labeled datasets. Our empirical results demonstrate that training diffusion models with style ambiguity loss significantly enhances their ability to generate novel outputs. These models consistently achieve higher scores on empirical metrics compared to models trained without style ambiguity loss. These creatively trained diffusion models outperform GANs trained with style ambiguity loss as well. Our findings suggest that incorporating style ambiguity can be a powerful approach to foster creativity and diversity in generated content, opening new avenues for future research in the field of generative models. Further work remains. As noted, the use of the CLIP alignment function for utility tends to product a lot of faces for the prompts we used. Use of another alignment technique may be better, or it could be supplemented with some sort of method to increase output diversity (Sadat et al., 2024). Furthermore, this method can likely be applied to different datasets, consisting of different images or even different modalities like audio or video. Using the K-Means Classifier for style ambiguity loss is particularly adept at this, given there is no need for *any* labels.

## Broader Impact Statement

Many are concerned about the impacts of generative AI. By making art, this work infringes upon a domain once exclusive to humans. Companies have faced scrutiny for possibly using AI (Gutierrez, 2024), and many creatives, such as screenwriters and actors, have voiced concerns about whether their jobs are safe (del Barco, 2023). Nonetheless, using AI can help humans by making them more efficient, providing inspiration, and generating ideas (Fortino, 2023; Campitiello, 2023; Darling, 2022). It's also not certain how copyright protection will function for AI-generated art (Watiktinnakorn et al., 2023), given copyright law is based on the premise that creative works originate solely from human authorship. Clear, consistent policies, both at the government level and by industry and/or academic groups, will be needed to mitigate the harm and maximize the benefits for all members of society.

# 7   Assistance

## Author Contributions

## Acknowledgements

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

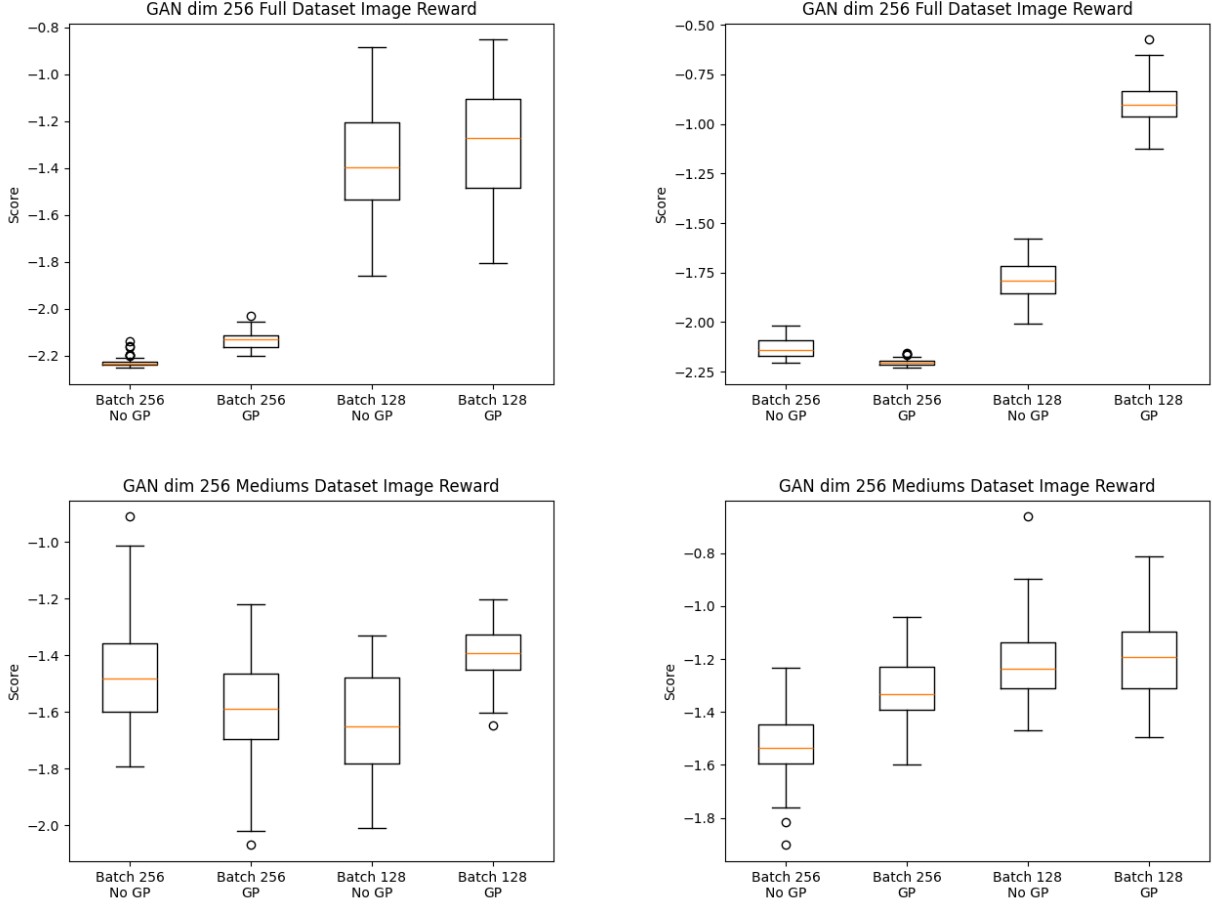

Table 8: Image Reward

# A  GAN Results

Box plots for the image reward and aesthetic scores are shown in tables 8 and 9, respectively. Some example images are shown in figure 4

# B  Additional Images

We display a few more images in figures 5, 6 and 7. Once again, each image in each row was generated with the same seed and same prompt.

# C  WikiArt Classes

We host the original **WikiArt-Full** dataset on `REDACTEDWHILEUNDERBLINDREVIEW`. The 27 WikiArt style classes, $L_{full}$ are listed in table 10

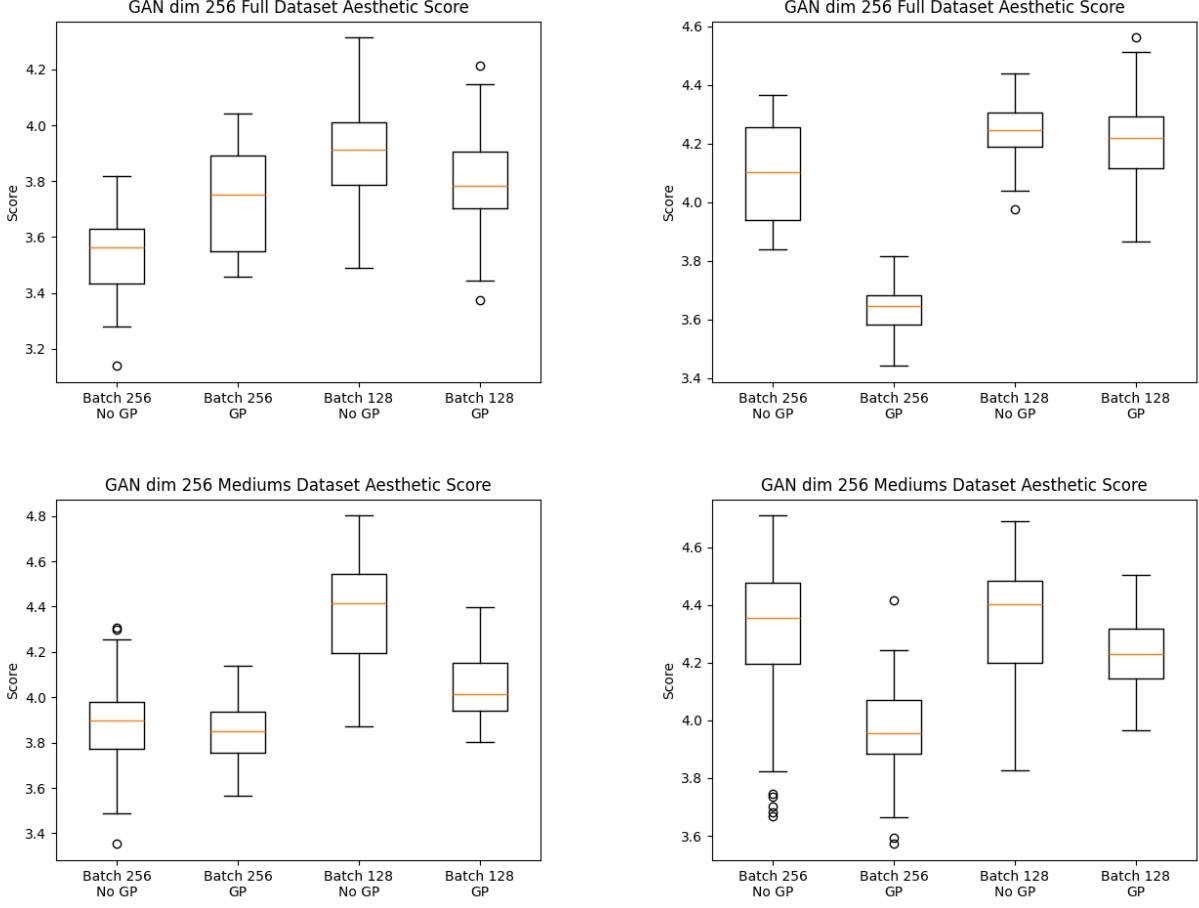

Table 9: Aesthetic Reward

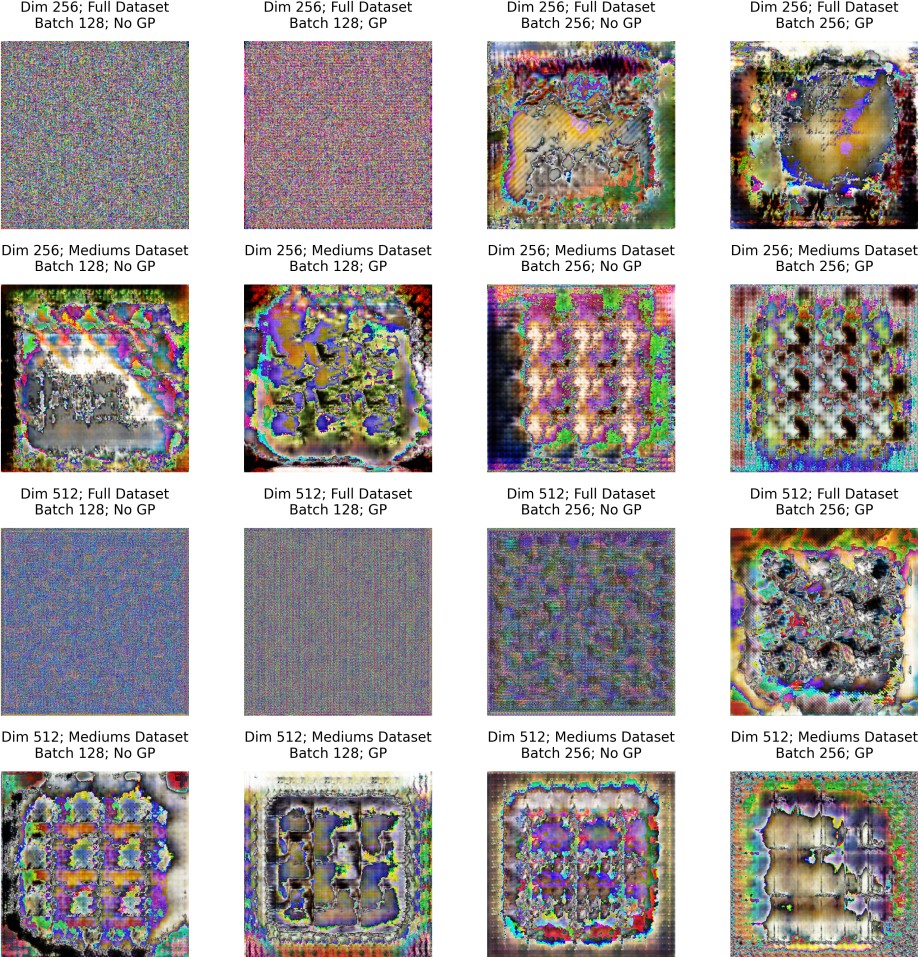

Figure 4: GAN Examples

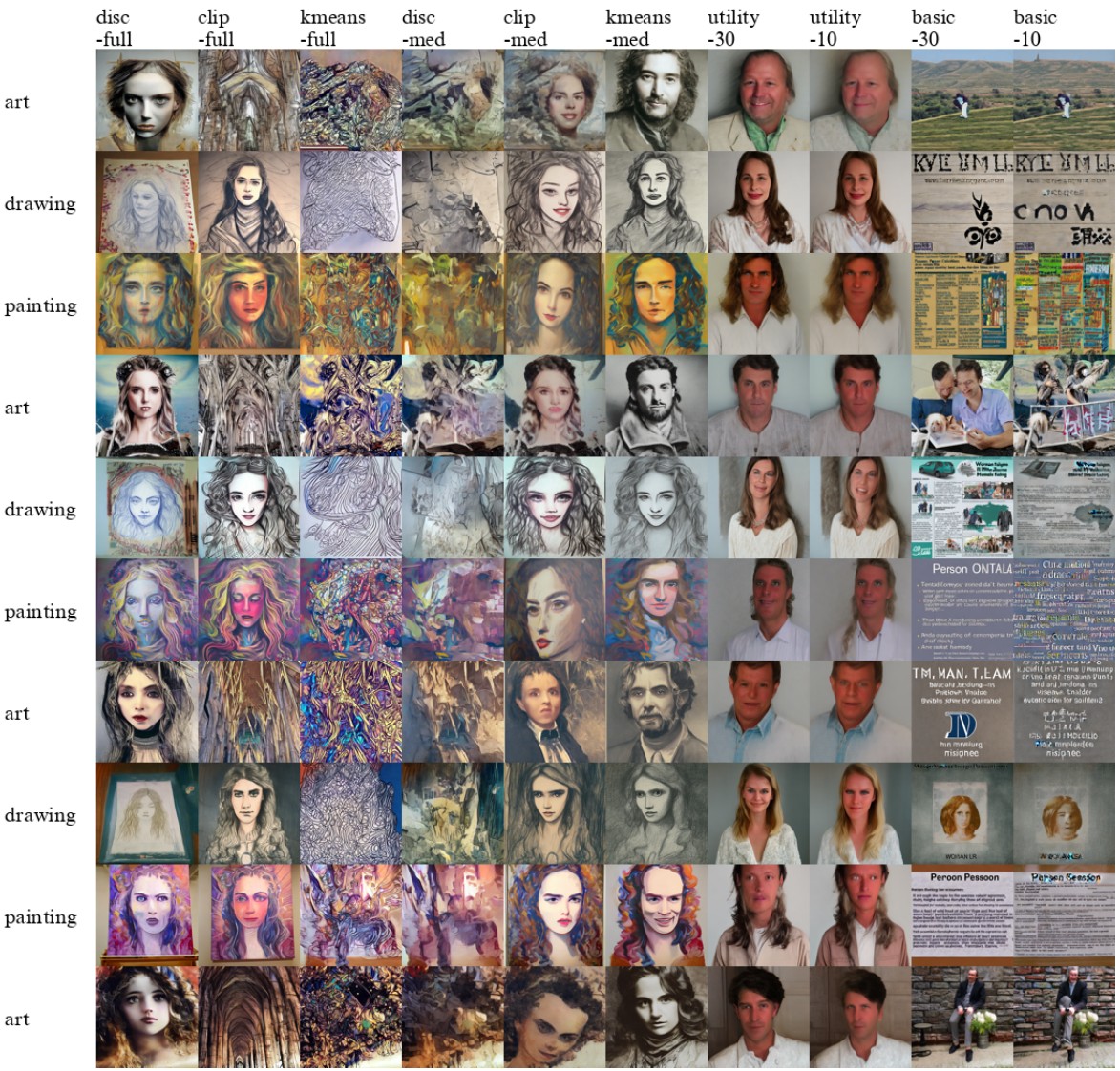

Figure 5: More Images

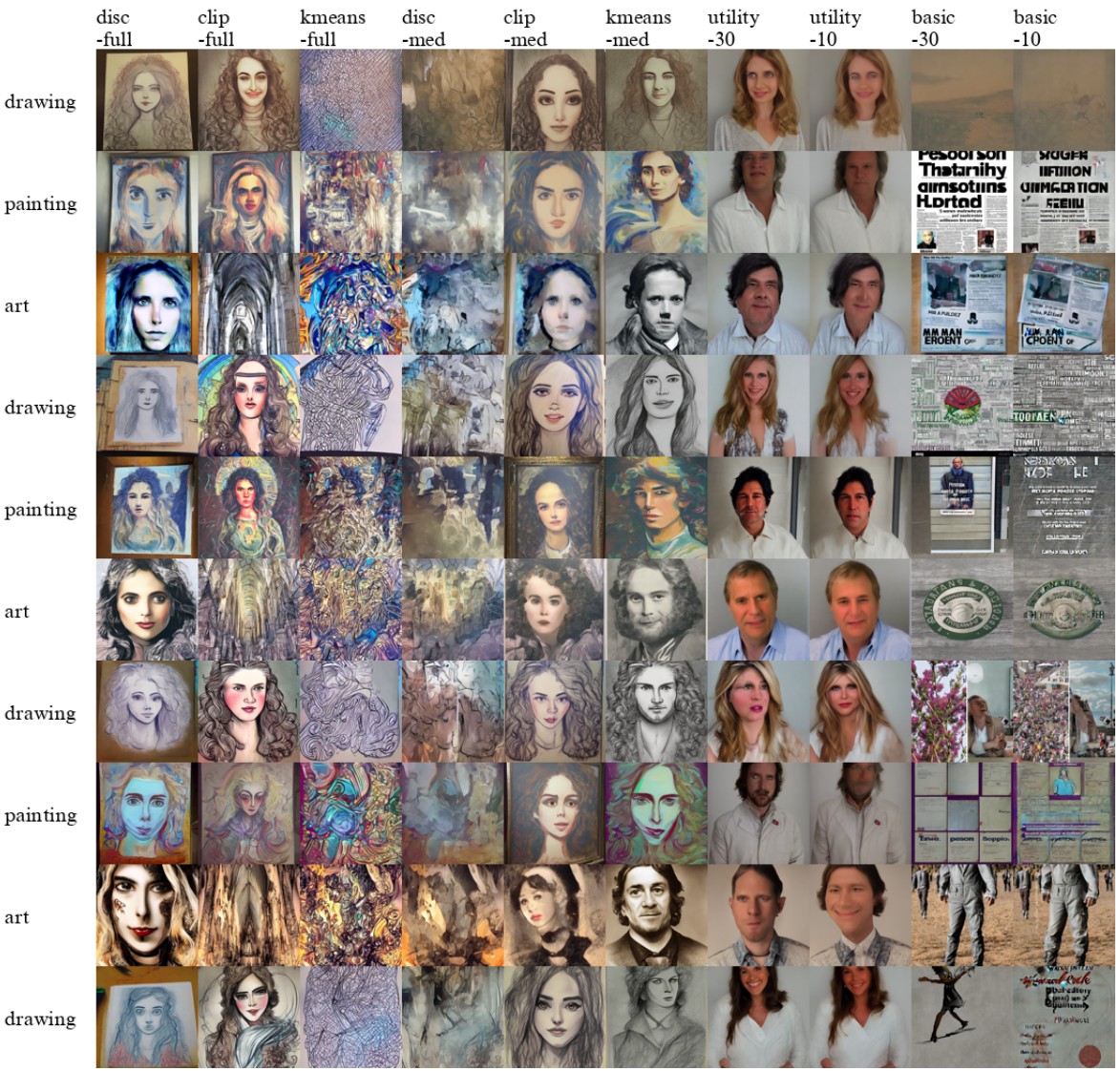

Figure 6: More Images

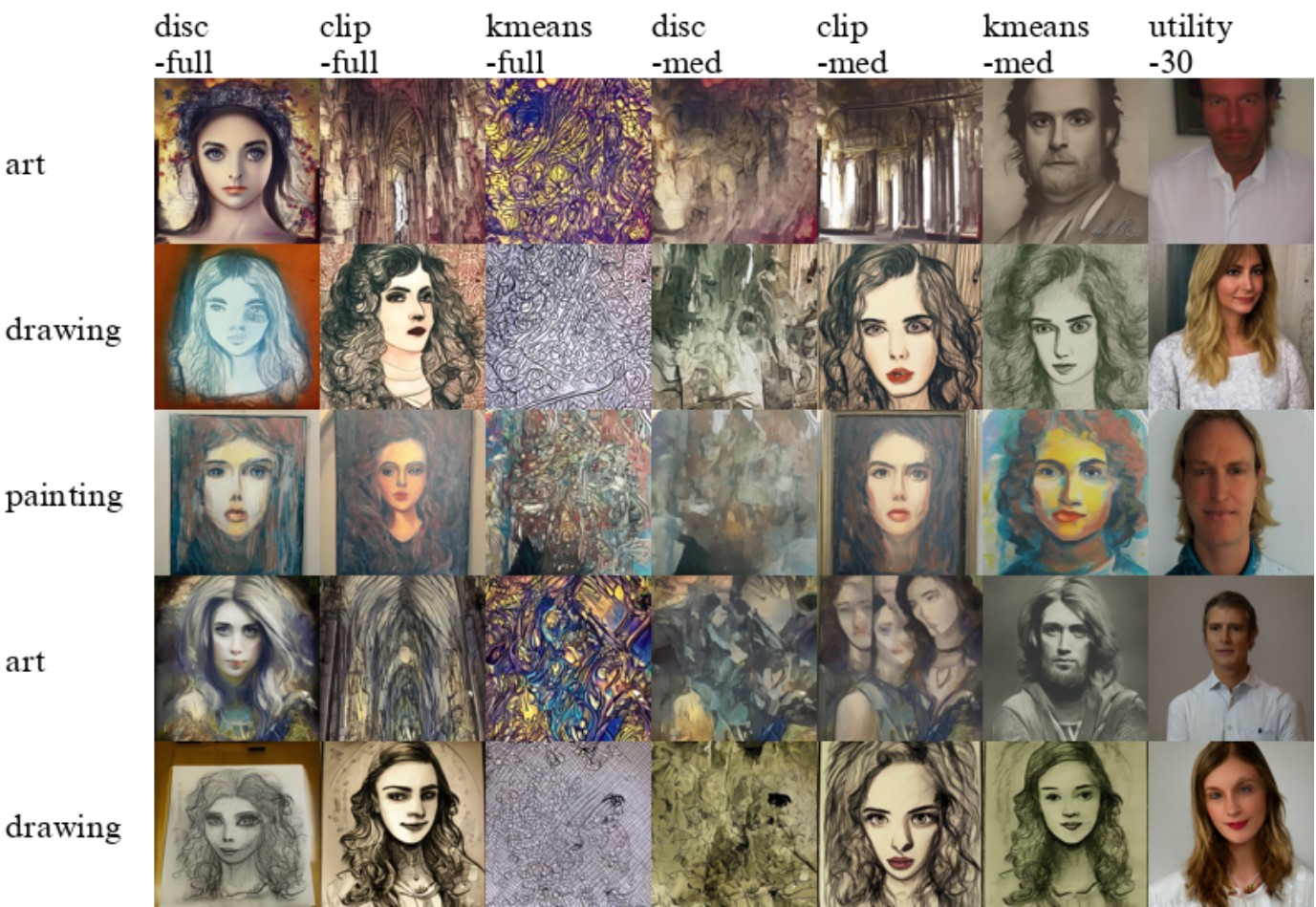

Figure 7: More Images

| contemporary-realism | art-nouveau-modern | abstract-expressionism |
|---|---|---|
| northern-renaissance | mannerism-late-renaissance | early-renaissance |
| realism | action-painting | color-field-painting |
| pop-art | new-realism | pointillism |
| expressionism | analytical-cubism | symbolism |
| fauvism | minimalism | cubism |
| romanticism | ukiyo-e | high-renaissance |
| synthetic-cubism | baroque | post-impressionism |
| impressionism | rococo | na-ve-art-primitivism |

Table 10: Styles

| style class | quantity | percentage |
|---|---|---|
| expressionism | 6054 | 91.56 |
| post-impressionism | 5832 | 89.25 |
| fauvism | 841 | 96.08 |
| abstract-expressionism | 2518 | 89.95 |
| na-ve-art-primitivism | 2148 | 93.39 |
| cubism | 2027 | 91.61 |
| synthetic-cubism | 197 | 89.85 |
| analytical-cubism | 105 | 91.43 |
| new-realism | 280 | 96.07 |
| action-painting | 93 | 93.55 |

Table 11: Mediums

# D  Medium Subset

## D.1  Captions

All captions were generated using the **Salesforce/blip-image-captioning-base** checkpoint downloaded from `https://huggingface.co/Salesforce/blip-image-captioning-base`.

## D.2  Subset

The categories of images that had the highest percentage of images in the categories with text captions that contained one of the words in the relevant text prompt set, as well as the percentage of images that did so, and the quantity of images in said category are shown in table 11 . We host the **Mediums** datasets at `REDACTEDWHILEUNDERBLINDREVIEW`.

# E  Training

For reproducibility and transparency, the hyperparameters are listed in table 12 and table 13. All experiments were implemented in Python, building the models in **pytorch** (Paszke et al., 2017) using **accelerate** (Gugger et al., 2022) for efficient training. The diffusion models also relied on the **trl** (von Werra et al., 2020), **diffusers** (von Platen et al., 2022) and **peft** (Mangrulkar et al., 2022) libraries. The K-Means clustering was done using the k means implementation from **scikit-learn** (Pedregosa et al., 2011). A repository containing all code can be found on github at `REDACTEDWHILEUNDERREVIEW`. Each experiment was run using two NVIDIA A100 GPUs with 40 GB RAM.

| Hyperparameter | Value |
|---|---|
| Epochs | 25 |
| Effective Batch Size | 8 |
| Batches per Epoch | 32 |
| Inference Steps per Image | 30 |
| LORA Matrix Rank | 4 |
| LORA $\alpha$ | 4 |
| Optimizer | AdamW |
| Learning Rate | 0.0015 |
| AdamW $\beta_1$ | 0.9 |
| AdamW $\beta_2$ | 0.99 |
| AdamW Weight decay | 1e-4 |
| AdamW $\epsilon$ | 1e-8 |

Table 12: DDPO Hyperparameters

| Hyperparameter | Value |
|---|---|
| Epochs | 100 |
| Batch Size | 32 |
| Optimizer | Adam |
| Learning Rate | 0.001 |
| Adam $\beta_1$ | 0.9 |
| Adam $\beta_2$ | 0.99 |
| Adam Weight decay | 0.0 |
| Adam $\epsilon$ | 1e-8 |
| Noise Dim | 100 |
| Wasserstein $\lambda$ | 10 |
| Leaky ReLU negative slope | 0.2 |
| Convolutional Kernel | 4 |
| Convolutional Stride | 2 |
| Transpose Convolutional Kernel | 4 |
| Transpose Convolutional Stride | 2 |

Table 13: CAN Hyperparameters

### E.1 Architecture

For diffusion model training, the text encoder, autoencoder and unet were all loaded from `https://huggingface.co/stabilityai/stable-diffusion-2-base`. These model components were all frozen, but we added trainable LoRA weights to the cross-attention layers of the Unet. Parameter counts are shown in table 14. The diffusion model components used the same amount of parameters regardless of image size, but the generator and discriminator had more parameters as image size increased.

| Model Component | Total Parameters | Trainable Parameters | Percent Trainable |
|---|---|---|---|
| Text Encoder | 34,0387,840 | 0 | 0% |
| Autoencoder | 83,653,863 | 0 | 0% |
| UNet | 866,740,676 | 829,952 | 0.1% |
| Generator (Image Dim 512) | 48,014,784 | 48,014,784 | 100% |
| Discriminator (Image Dim 512) | 20,115,932 | 20,115,932 | 100% |

Table 14: Parameter Counts

We used the convolutional neural network (Dumoulin & Visin, 2018) architecture described in Elgammal et al. (2017) for the CAN but had to use more/less layers to produce higher/lower dimension images. The generator takes a $1 \times 100$ gaussian noise vector $\in \mathbb{R}^{100} \sim \mathcal{N}(0, I)$ and maps it to a $4 \times 4 \times 2048$ latent space, via a convolutional transpose layer with kernel size $= 4$ and stride $=1$, followed by 6 transpose convolutional layers each upscaling the height and width dimensions by two, and halving the channel dimension (for example one of these transpose convolutional layers would map $\mathbb{R}^{4 \times 4 \times 2048} \to \mathbb{R}^{8 \times 8 \times 1024}$) followed by batch normalization (Ioffe & Szegedy, 2015) and Leaky ReLU (Maas et al., 2013), and then one final convolutional transpose layer with output channels $= 3$ and tanh (Dubey et al., 2022) activation function. Diagrams of the generator is shown in the figure 8.

For the discriminator, we first applied a convolution layer to downscale the input image height width dimensions by 2 and mapped the 3 input channel dimensions to 32 ($\mathbb{R}^{512 \times 512 \times 3} \to \mathbb{R}^{256 \times 256 \times 32}$) with Leaky ReLU activation. Then we had 5 convolutional layers each downscaling the height and width dimensions by 2 and doubling the channel dimension (for example, one of these convolutional layers would map $\mathbb{R}^{256 \times 256 \times 32} \to \mathbb{R}^{128 \times 128 \times 64}$) with batch normalization and Leaky ReLU activation. Then we had two more convolutional layers, each downscaling the height and width dimensions but keeping the channel dimensions constant (using the prior layer's channel dimensions), with batch normalization and Leaky ReLU activation. The output of the convolutional layers was then flattened. The discriminator had two heads- one for style classification (determining which style a real image belongs to) and one for binary classification (determining whether an image was real or fake). The binary classification head consisted of one linear layer with one output neuron. The style classification layer consisted of 2 linear layers with LeakyReLU activation and Dropout, with output 1024 output neurons and 512 output neurons, respectively, followed by a linear layer with 27 output neurons for the 27 artistic style classes. A diagram of the discriminator is shown in 9.

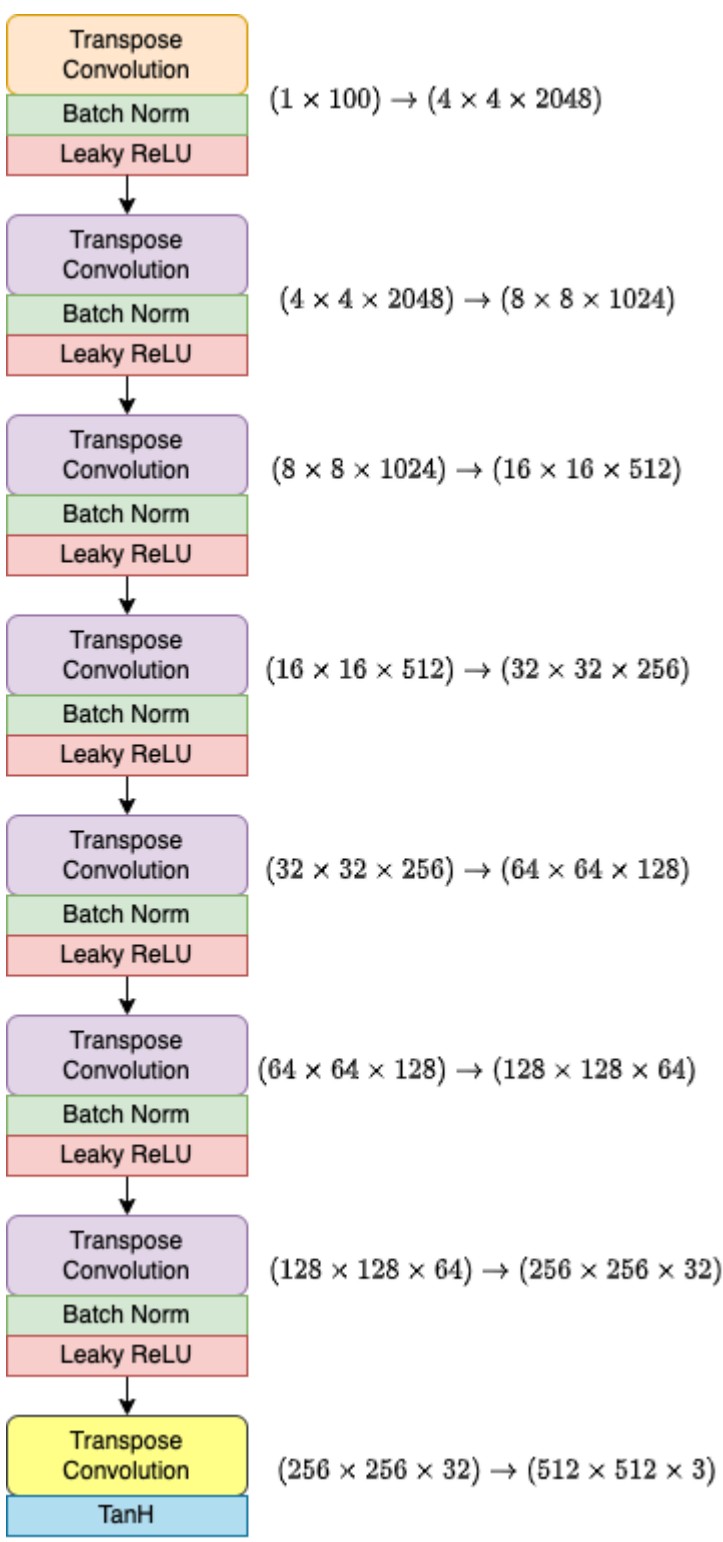

Figure 8: Generator Architecture (Image Dim 512)

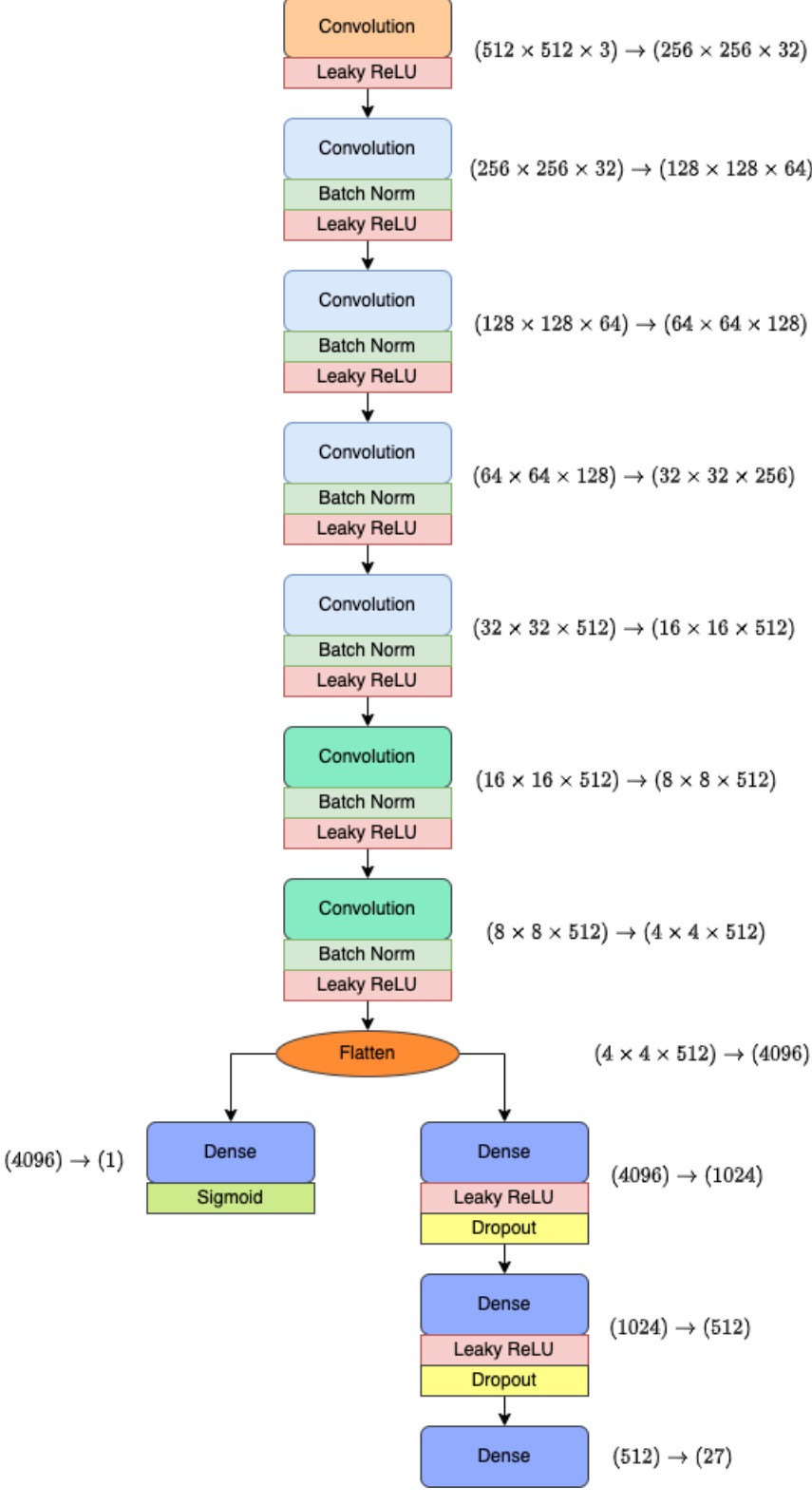

Figure 9: Discriminator Architecture (Image Dim 512)

