# OpenReview forum: "Using Style Ambiguity Loss to Improve Aesthetics of Diffusion Models"
_TMLR — Rejected by TMLR_

### Review · Reviewer_4w27 · 2024-08-23

**Summary Of Contributions:**

This paper proposes to apply the style ambiguity loss proposed by Creative Adversarial Network to diffusion models via denoising diffusion proximal optimization (DDPO). Specifically, a diffusion model is optimized via DDPO such that generated images maximize the classification loss of a style classifier. The authors propose three variants of the ambiguity loss -- ambiguity loss with a discriminator, a CLIP-based classifier, and a K-means classifier in the latent space of a CLIP encoder.

**Audience:**

Yes

**Broader Impact Concerns:**

No broader impact concerns.

**Claims And Evidence:**

No

**Requested Changes:**

Below, I list suggestions for addressing the Weaknesses mentioned in the previous section.

- To address [W1], I suggest the authors apply the same set of experiments in [1] to their model. Running even a subset of the experiments in [1] would be very helpful. This is critical to securing my recommendation.
- To address [W2], I suggest the authors compare the generated sample quality and novelty of CAN as well. This should be relatively simple, as pre-trained models are available at [2]. This would strengthen the work in my opinion.
- To address [W3], the authors could use a more meaningful notation for the models. For instance, the model trained with a discriminator, $\lambda_{novelty}=1$ and $\lambda_{utility} = 0.25$ could be denoted something along the lines of $disc_{1,0.25}$. This would also strengthen the work

[1] CAN: Creative Adversarial Networks

[2] https://github.com/mlberkeley/Creative-Adversarial-Networks

**Strengths And Weaknesses:**

**Strengths**

- [S1] The paper is clearly written and the method is intuitive and easy to understand.
- [S2] Effects of the proposed techniques on AVA and IR are examined with a thorough ablation study.

**Weaknesses**

- [W1] The metrics used to evaluate the model, AVA and IR, measure image quality, but not novelty. Hence, it is difficult to verify whether the proposed method truly generates novel images. This undermines the authors' claim that the model is capable of generating novel and creative data.
- [W2] The paper lacks comparison with previous works, such as the Creative Adversarial Network.
- [W3] The experiment section is slightly confusing, as the names for the model M0, ..., M9 do not indicate the training configuration. I had to keep going back and forth the results and Table 1.

---

> ### Author Response · Authors · 2024-08-23
> **Response asking for clarification**
>
> Thank you for your feedback! When you reference "the same set of experiments in [1]", I assume you mean experiments I,II,III and IV in the original Creative Adversarial Network paper by Elgammal et al, which were a set of user studies asking human subjects what they thought of the work. Is that correct?

---

> > ### Comment · Reviewer_4w27 · 2024-08-27
> >
> > Yes, that is correct.

---

> ### Author Response · Authors · 2024-08-29
> **Regarding the pre-trained models**
>
> The repo @ https://github.com/mlberkeley/Creative-Adversarial-Networks no longer has the pretrained models (the models are supposedly in a google drive folder that is linked, but said folder is empty). While I can use my own trained CAN to generate samples. However, these samples are very low quality compared to those of the diffusion model, due to the inherent instability of training GANs (https://link.springer.com/article/10.1007/s11042-024-19361-y) .

---

> ### Author Response · Authors · 2024-09-10
> **Update**
>
> I have addressed your criticisms in the newest revision

---

### Review · Reviewer_u3hT · 2024-10-05

**Summary Of Contributions:**

This papser proposes to improve the "creativity" of diffusion model using the style ambiguity loss in training. Empirically they show that the generated imagery results are better when comparing to the results of either vanila diffusion models or GANs.
Main contributions include:
*apply style ambiguity loss to diffusion models via reinforcement learning
*develope versatile CLIP-based and K-Means-based creative style ambiguity losses
*show the generated imagery results are better when comparing to the results of either vanila diffusion models or GANs.

**Audience:**

No

**Broader Impact Concerns:**

The work has limited impact in its current form. Too few empirical results and limited tech contributions.

**Claims And Evidence:**

No

**Requested Changes:**

*strengthen the technical contribution
*complete the missing parts of the sentences
*show a lot more empirical results
*creativity is very subjective. How to empirically evaluate & claim one is more creative than the other? It is not clear to me.

**Strengths And Weaknesses:**

The goal of generating creative results are grant and interesting. Meanwhile, there are a number of major issues with the paper. The research story presented in the paper seems to be far from ready to be published at TMLR, even after possible major revisions.
* Limited technical contributions. The technical contributions are rather limited. For example, only 1.5 pages are used to describe the technical details of the proposed method, including datasets, labels, the RL reward function, and the two types of losses:  CLIP-based and K-Means-based style ambiguity losses. Not sure which one of them is new. Overall the paper is lacking in terms of technical details and contributions.
* Limited empirical evaluation results. The proposed method seems very limited in the visuals and the number & diversity of generated images, as shown in Figs. 1&2. Besides, the contents in the Figures 1&2 are not well explained.
* Presentation could be improved. In terms of presentation, some part of the paper seems not well written. For example, the sentences right before Sec. 4.3 & Sec. 4.4 are not completed.
* I am confused that in the title it says to improve the "Aesthetics" but ever since the abstract, the goal is entirely about to improve the "creativity", with no mentioned at all of the "Aesthetics". I am not convinced that these two words have the same meaning.

---

> ### Author Response · Authors · 2024-10-06
> **Response**
>
> Thank you for your review. A few questions asking for clarification:
> 1. By "strengthen the technical contribution" what do you mean by that? Is it unclear what the CLIP-Based and K-Means classifiers are? I concede they are not complicated and do not require that much explanation, but even "simple " solutions can be effective (classifier-free guidance, for example, is wondrously simple, and the authors of the original paper devote roughly a page to explaining their new method https://arxiv.org/abs/2207.12598).
> 2. You asked for  us to "show a lot more empirical results". What other metrics would you suggest we use?

---

> ### Comment · Reviewer_u3hT · 2024-11-03
>
> Summary Of Contributions:
>
> This paper proposes to improve the "creativity" of diffusion model using the style ambiguity loss in training. Empirically they show that the generated imagery results are better when comparing to the results of either vanila diffusion models or GANs. Main contributions include: *apply style ambiguity loss to diffusion models via reinforcement learning *develope versatile CLIP-based and K-Means-based creative style ambiguity losses *show the generated imagery results are better when comparing to the results of either vanila diffusion models or GANs.
>
> Strengths And Weaknesses:
>
> The goal of generating creative results are grant and interesting. Meanwhile, there are a number of major issues with the paper. The research story presented in the paper seems to be far from ready to be published at TMLR, even after possible major revisions.
>
> Limited technical contributions. The technical contributions are rather limited. For example, only 1.5 pages are used to describe the technical details of the proposed method, including datasets, labels, the RL reward function, and the two types of losses: CLIP-based and K-Means-based style ambiguity losses. Not sure which one of them is new. Overall the paper is lacking in terms of technical details and contributions.
>
> Limited empirical evaluation results. The proposed method seems very limited in the visuals and the number & diversity of generated images, as shown in Figs. 1&2. Besides, the contents in the Figures 1&2 are not well explained.
>
> Presentation could be improved. In terms of presentation, some part of the paper seems not well written. For example, the sentences right before Sec. 4.3 & Sec. 4.4 are not completed.
>
> I am confused that in the title it says to improve the "Aesthetics" but ever since the abstract, the goal is entirely about to improve the "creativity", with no mentioned at all of the "Aesthetics". I am not convinced that these two words have the same meaning.

---

> > ### Author Response · Authors · 2024-11-04
> > **Possible Error**
> >
> > This was a copy and paste of your original review. Are you sure this was intentional?

---

### Review · Reviewer_UvvL · 2024-10-16

**Summary Of Contributions:**

This paper proposes to improve creativity and aesthetics of diffusion image generative models with the ambiguity loss.  The diffusion model is trained with reinforcement learning, where the reward is characterized by classifier-, CLIP, and clustering-based ambiguity loss.  The paper shows that the ambiguity loss guides the diffusion model to generate images closer to human preference and better novelty.

**Audience:**

Yes

**Broader Impact Concerns:**

The idea of improving creativity of image generative models might lead to the generation of inappropriate contents, which may involve issues of sex, races and violence.  As those contents are rarely seen in the existing dataset, generating these contents could be considered novel according to the proposed method.  The authors should discuss this concern in the Broader Impact Statement.

**Claims And Evidence:**

Yes

**Requested Changes:**

1. $N_s$ in page 2 should be clearly defined
2. $C$ in $\mathcal{L}_{CA}$ should be defined
3. The formulation of DDPO should be provided, otherwise, readers might not know the context
4. The definition of $\nabla_\theta \mathcal{J}_{DDRL}$ should be provided
5. In Eqn of section 4.4.3, $KMEANS(x_0)$ should be changed to $KM(x_0)$ not $KMEANS(x_0)$, as the former is not defined
6. The caption of Table 4 should be provided
7. The user study scores should involve CAN
8. The experimental design of the user study should be re-considered

**Strengths And Weaknesses:**

Strengths:
1. The proposition of this paper is clearly illustrated in the introduction.  Readers can easily catch the high-level idea and objective of this paper easily.
2. The proposed method that define rewards with the ambiguity loss is technically sound.

---

Weaknesses:
1. Some notations are not clearly defined.
2. This paper does not provide the formulation of some prior works, which the proposed method is heavily based on.
3. The user study does not involve an important baseline--CAN (Elgammal et al., 2017)
4. The design of the user study is questionable.  From Figure 1, 2, 5, 6 and 7, I cannot tell if *disc-full*, *clip-full*, *kmeans-full*, *disc-med*, *clip-med* or *kmeans-med* is better than the others.  The primary reason is that I am not sure if the generated image is novel from one-to-one comparison.  In fact, I'd argue that we can only know if the generated image is novel by comparing against many other images of the same topic/content.  On the other hand, I'd consider an image novel if it is sampled from different styles from all the others.  Since the primary goal of this paper is to increase the creativity/novelty of image generative models, I believe the authors should come up with a more solid experimental design for user study to support their proposition.
5. Have you consider to use the instance-discrimination loss (e.g. MOCO / DINO / DINO.v2) to characterize the ambiguity loss?  An image could be novel if it doesn't look like any other images.

---

### Review · Reviewer_JXf8 · 2024-10-16

**Summary Of Contributions:**

This paper proposes to use style ambiguity loss to train a diffusion model and find that this loss can generate better images than the baseline diffusion models and GANs. However, the style ambiguity loss is not new. This limits the contribution of this work.

**Audience:**

Yes

**Broader Impact Concerns:**

No broader impact concerns.

**Claims And Evidence:**

No

**Requested Changes:**

Please include more related work in the diffusion models with loss guidance.

Need more technical contributions for a solid paper.

**Strengths And Weaknesses:**

Strengths:

The idea of applying style ambiguity loss to diffusion models to achieve creativity in generated images is easy to follow.

Weaknesses:

The paper's core contribution—transferring the style ambiguity loss from GANs to diffusion models—is fundamentally narrow. Applying an existing concept from one generative model to another does not constitute a significant scientific advancement unless substantial new insights or innovations accompany it.

The paper misses substantial related work, especially in the realm of diffusion models with loss guidance. Many recent advancements and techniques in diffusion models that could be relevant to this research have not been discussed or even mentioned.

The manuscript reads more like an experimental report than a structured technical paper. There is a heavy emphasis on the application and empirical findings with insufficient theoretical underpinning or novel algorithmic development.

The visualization results do not show promising results compared to existing methods.

There are also some minor issues, such as the format of Table 6,7.

---

### Decision · Action_Editor_rxyw · 2024-12-10

**Recommendation:** Reject

**Comment:**

The paper received mixed evaluations from four reviewers. While they acknowledged that the paper addresses an interesting problem and presents an easy-to-follow strategy, they also identified several significant concerns, as detailed below:
1. Limited Technical Contribution (JXf8, u3hT, 4w27):
    * The proposed solution combines the ambiguity loss from CAN and a diffusion model trained with DDPO. The reviewers found this approach to be straightforward and incremental.
2. Lack of Proper Evaluation Metrics for Image Novelty (UvvL, u3hT, 4w27):
    * The evaluation metrics used focus on measuring image quality rather than novelty. Additionally, the qualitative results provided are insufficient to demonstrate the method's effectiveness in generating novel images.
3. Insufficient Experimental Comparisons/Results (UvvL, u3hT, 4w27):
    * The experimental evaluation lacks comparisons with previous works, such as the Creative Adversarial Network, making it difficult to assess the proposed method's advancements.
4. Lack of Clarity in Presentation and Poor Writing (JXf8, UvvL, u3hT, 4w27):
    * The paper contains multiple instances of unexplained notations, formatting issues, and incomplete sentences, which hinder clarity and comprehension.

The authors partially addressed some of these concerns, particularly point 3. However, they did not fully resolve the reviewers' questions. During the discussion phase, one reviewer felt that the major concerns were mostly addressed and leaned towards acceptance. In contrast, the other reviewers remained unconvinced due to the issues mentioned above and recommended rejection.

Based on the revised draft, reviews, and discussion, the AE finds the majority of the reviewers' arguments compelling and sees no sufficient reason to overturn their ratings. Overall, the effectiveness of the proposed strategy remains unclear, and the paper requires significant revision to improve its presentation before it is ready for publication in TMLR. The authors are encouraged to revise the paper accordingly and consider resubmitting at a later time.

**Audience:**

The paper addresses a challenging problem in controlled image generation regarding creativity, which should be interesting to the audience in generative visual models.

**Claims And Evidence:**

The paper proposes to improve the creativity of artificially generated images via diffusion models trained with the ambiguity loss proposed by Creative Adversarial Network (CAN).  To achieve this, it adopts denoising diffusion proximal optimization (DDPO) for model training. and develops three variants of the ambiguity loss that use a neural network classifier, a CLIP-based classifier, and a K-means classifier in the CLIP latent space, respectively.

The main claim of this work lies in the effectiveness of this integrated strategy regarding the creativity of generated images.  While the paper presents an easy-to-follow solution for a challenging problem, it falls short in providing convincing evidence due to lacking proper evaluation metrics for the image novelty, missing important baselines in its empirical study and unclear presentation.

**Resubmission Of Major Revision:**

The authors may consider submitting a major revision at a later time.